# Effects of Different Kinds of Fruit Juice on Flavor Quality and Hypoglycemic Activity of Black Tea

**DOI:** 10.3390/foods14040588

**Published:** 2025-02-10

**Authors:** Hongchun Cui, Yuxiao Mao, Yun Zhao, Weihong Huang, Jianyong Zhang

**Affiliations:** 1Tea Research Institute, Hangzhou Academy of Agricultural Science, Hangzhou 310024, China; chc1134@126.com (H.C.);; 2Tea Research Institute, Chinese Academy of Agricultural Science, Hangzhou 310008, China; 3Zhejiang Agricultural Technology Extension Center, Hangzhou 310024, China

**Keywords:** black tea, pear, apple, polyphenol oxidase, flavor quality, modulation, α-amylase inhibition, hypoglycemic activity

## Abstract

At present, the heavy bitter taste, poor flavor quality and low functional activity of summer and autumn tea are the bottleneck problems restricting the low utilization rate of summer and autumn tea resources. The research and development of new products of fruit-flavored black tea is conducive to expanding the utilization of summer and autumn tea resources. Different kinds of fruit juice were added during the fermentation and processing of classic black tea, such as bananas, apples, fragrant pear and Sydney pear, in this study. The effects of fruit juice on the flavor quality and amylase inhibitory activity of fruity black tea were researched. The sensory quality, flavor chemicals and α-amylase inhibitory activity were evaluated. The results showed that the sensory evaluation scores of black tea treated with fruit juice were significantly higher than those of black tea treated without fruit juice, especially the crown pear juice. The amylase inhibition rate of black tea treated with fruit juice was significantly higher than the control treated without fruit juice (*p* < 0.05). The sensory evaluation scores, polyphenol oxidase activity, water extract content, soluble sugar content, free amino acid content, theaflavin content, thearubigin content and inhibition rate of amylase activity of black tea treated with pear juice were significantly higher than those of the apple and banana juices (*p* < 0.05), especially crown pear juice. Tea polyphenol content and theaflavin content of black tea treated with added pear juice were significantly lower (*p* < 0.05) than the black tea control treated with added apple juice and banana juice, especially crown pear juice. The fruity black tea treated with crown pear juice had a redder broth, more pronounced sweet fruit aroma, sweet and mellow taste and reduced astringency. Therefore, the black tea treated with crown pear juice was preferred. The research hopes to provide a theoretical basis for the research of black tea quality control and the research of summer and autumn tea resources utilization technology.

## 1. Introduction

Black tea is a global health drink with special flavor characteristics, loved by people all over the world. Black tea is also an important advantageous agricultural product in China, which plays an important role in China’s agricultural production, local economy and export earnings. In recent years, China’s tea plantation area and tea production have steadily increased. However, China’s tea consumption has been growing slowly. The possible reason was that more than 80% of China’s tea gardens were not effectively utilized in the summer and fall for fresh tea leaves [1,2]. In 2023, the tea plantation area in China exceeded 50 million acres. The tea production in China was 3.55 million tons. With the continuous increase in China’s tea plantation area, the contradiction of structural production over sales was becoming more and more prominent [3]. A large number of summer and fall tea leaves were abandoned. The lots of summer and fall tea resources were wasted, which inducted a direct economic loss of more than EUR 10.6 billion. It was an important issue to be solved for the sustainable development of China’s tea industry [4]. Abundant tea polyphenols existed in the fresh leaves of summer and autumn tea [5], which was an important breakthrough to expand the utilization of summer and autumn tea resources.

In recent years, China’s industrial tea beverage and new-style tea beverage industries developed rapidly by activating the new-style consumption of tea beverages by young groups. The tea beverage industry had an urgent demand for raw tea to highlight flavor quality, especially fruit and fruit-flavored raw tea. The research and development of new black tea, new oolong tea and other processing technologies continued to deepen. It became the key breakthrough in solving the problem of the high-value utilization of tea resources in summer and autumn in China. The faster the metabolic rate of tea tree, the more bitter and astringent the flavor and poor flavor quality of the black tea [5]. The main reason is that the accumulation of ester catechins, caffeine, anthocyanins and other bitter ingredients in fresh tea leaves in summer and autumn is higher than that in fresh tea leaves in spring, which is due to the high temperature in summer and autumn [6].

The controlling of oxygen level [7], controlling of fermentation humidity [8], cellaring flowers [9], controlling of fermentation time [10,11,12], controlling of fermentation temperature [13,14], and addition of exogenous enzymes [15] were mainly used to improve the flavor quality and health function activity of tea. However, summer and fall black teas still have more pronounced bitter and astringent flavors and lower health functional activities. The black tea blending method was widely favored by consumers in various countries around the world. The blend of flowers or fruits enhanced the flavor of black tea. Fruits from different sources were sweet and refreshing, which can increase the sweet aroma of black tea [16,17,18,19,20]. The oxidoreductase of fruit and tea could form together to promote the conversion of polyphenols [21,22,23,24,25]. Moreover, fruit could increase the content of functional components of black tea and then improve the biological activity of black tea [26,27,28]. There were abundant fruit resources, such as pear, apple, banana and lemon, in addition to rich tea resources in China. Fruity black tea could not only improve the flavor and function of black tea but also make full use of the excess capacity of fruit and summer and autumn tea resources.

Black tea not only had unique flavor properties, but also had a variety of functional activities, such as tea polyphenols, theaflavins, thearubigins, theobromines, caffeine, tea polysaccharides, etc., [29,30]. The significant hypoglycemic effects and the prevention and treatment of type 2 diabetes mellitus (T2DM) of these components had been demonstrated [31,32,33]. There were relatively few studies on the hypoglycemic activity of black tea with different types of fruit juices. There was an urgent need to preliminarily investigate the hypoglycemic activity of fruit-flavored black tea by inhibiting the activity of α-amylase. According to the existing studies and the basis of our previous experiments, it is speculated that fruit-flavored black tea may regulate hypoglycemic activity by regulating the oxidative polymerization of tea polyphenol.

In response to the problems of heavy bitter and astringent taste, poor flavor quality and the low functional activity of summer and autumn tea, relatively inexpensive apples and pears were introduced into black tea processing. The principles of fruit–tea composite fermentation and fruit–tea flavor fusion were applied to reduce the bitter and astringent taste of black tea. In addition, the α-amylase inhibition activity of fruit-flavored black tea treated with different kinds of pear juice and apple juice were compared and analyzed. The hypoglycemic activity of different kinds of fruit-flavored black tea were explored. We hope to provide a theoretical basis for the research of black tea quality control and summer and autumn tea resources utilization. The research also hopes to be conducive to enriching the high value and diversified use of tea resources.

## 2. Materials and Methods

### 2.1. Materials

Fresh leaves of tea planted in Longjing population species in summer and autumn were taken from tea gardens in Hangzhou, Zhejiang Province, China. The methanol and acetonitrile (chromatographic purity grade) were purchased from Beijing Dingguo Biotechnology Co., Ltd. (Beijing, China). The dinitrosalicylic acid, sodium dihydrogen phosphate, folinol, potassium diphosphate, sodium potassium tartrate, ninhydrin and polyvinyl pyrrolidone (analytical pure) were purchased from Shanghai Aladdin Biotechnology Co., Ltd. (Shanghai, China). The banana, apple, fragrant pear, Sydney pear and crown pear were purchased from Hangzhou Xianfeng Co., Ltd. (Hangzhou, China).

### 2.2. Instruments

The UV-3600 UV–Visible spectrophotometer was purchased from Shimadzu Corporation, Japan (Shimadzu, Japan). The pH tester and electronic balance were purchased from Sartorius Scientific Instruments (Beijing) Co., Ltd. (Beijing, China). The Ultrasonic cleaner was obtained from Kunshan Shumei Experiment Equipment Co., Ltd. (Kunshan, China). The tissue grinder was obtained from Zhejiang Meibi Experiment Equipment Co., Ltd. (Jiaxing, China).

### 2.3. Preparation of Fresh Fruit Juice Containing Polyphenol Oxidase (PPO)

Clean bananas, apples, fragrant pear, Sydney pear and crown pear were peeled and homogenized. Each fruit was homogenized separately. Each homogenized juice was mixed with 1% PVPP citrate-dipotassium phosphate buffer (pH 6.0, 100 mM) in a 2:1 ratio separately. The crude enzyme solution of each fruit was obtained separately.

### 2.4. Determination of PPO Activity

First, 0.5 mL of the crude enzyme solution of banana, apple, fragrant pear, Sydney pear and crown pear was put into the different test tubes separately. The 3.9 mL phosphate buffer (pH 5.5) at a concentration of 0.05 mol·L^−1^ and 1.0 mL catechol at the a concentration of 0.1 mol/L were added into a test tube. The test tubes were maintained for 10 min at 25 °C in a thermostatic bath. The absorbance was measured at 525 nm by time scanning. A change in absorbance of 0.01 within 1 min was defined as one unit of enzyme activity (U).

### 2.5. Processing of Fruity Black Tea

One bud and two leaves of fresh summer and autumn tea were picked. The tea leaves were put at room temperature for one night and then put in the withering tank at 35 °C for withering. During the withering process, the leaves were turned every half an hour until the moisture content of the withered leaves was 60~62%. The withered leaves were rolled for 15 min by the kneading machine (Zhejiang Wuyi Zengrong food Machinery Co., Ltd., Wuyi, China). The way to roll leaves is as follows: light rolling for 3 min, heavy rolling for 8 min and light rolling for 4 min. The rolling leaves were sprayed with 10% (*v*/*w*) of each kind of fruit juice separately. Then, the tea leaves were fermented by a thermostat (Shanghai Yuming Instrument Co., Ltd., Shanghai, China) at 35 °C for 6 h. Finally, the tea leaves were thoroughly dried in a drying machine at 35 °C for 6 h (Zhejiang Shangyang Tea Machinery Co., Ltd., Quzhou, China). The parameters of the initial drying treatment were 105–110 °C temperature for 20 min. The parameter of the redrying treatment was redrying at 80–90 °C for 60 min. The fruit-flavored black teas with different flavor characteristics were successfully processed and obtained. Each treatment was replicated three times.

### 2.6. Sensory Evaluation of Fruity Black Tea

According to the China National Standard (GB/T 23776–2018) [34], the flavor quality of black teas was evaluated. Samples of 3.0 g fruity black tea were weighed separately, such as banana fruity black tea, apple fruity black tea, fragrant pear fruity black tea, Sydney pear fruity black tea and crown pear fruity black tea. The weighed black tea was put into a white porcelain cup with a capacity of 150 mL. Then, 150 mL of boiling water was added to the cup and the tea steeped for 5 min. An evaluation team was formed by 3 male and 2 female professional tea tasters. The panelists were instructed to smell and drink fruity black tea infusions and pause for 40 s between samples. The qualitative and quantitative assessment of the quality characteristics of the fruity black tea were carried out using the crypto sensory evaluation method.

A percentage system was used to evaluate the sensory quality of each fruity black tea sample. The fruity black tea was assigned 25% for tea appearance, 10% for tea infusion color, 25% for infusion aroma, 30% for tea infusion taste and 10% for tea residue. The 0.6 g black tea samples were weighed separately and placed in a 240 mL tea bowl. Then, 150 mL of boiling water was put into the tea bowl separately. After steeping for 3 min, the black tea liquor was stirred. The liquor color, aroma and taste of black tea samples were evaluated by 5 professional tea tasters in turn.

The total sensory score of black tea samples were calculated according to Equation (1):(1)Total sensory score=A×25%+B×10%+C×25%+D×30%+E×10%
where *A* refers to the tea appearance score, *B* refers to the tea infusion color score, *C* refers to the tea infusion aroma score, *D* refers to the tea infusion taste score and *E* refers to the tea residue score.

### 2.7. Inhibition of α-Amylase Activity by Fruity Black Tea Samples

Fruity black tea broth from Section 2.4 was used as the target material. Here, 0.2 mL of different concentrations of fruity black tea broth were mixed with 0.1 mL of α-amylase (0.5 U/mL) in a 2 mL centrifuge tube. The mixed mixtures were placed in incubation at 25 °C for 10 min. Subsequently, 0.2 mL starch (10 mg/mL) was added and further incubated for 10 min. Then, it was heated at 100 °C for 10 min. After cooling, 0.1 mL of the reaction solution was transferred to a 96-well enzyme labeling plate (0.1 M Phosphate buffer, pH 6.9). Absorbance at 540 nm was quantified using enzyme markers. The α-amylase inhibitory activity of each fruity black tea sample was obtained by calculation. The magnitude of α-amylase inhibitory rate was calculated as follows:(2)$$\alpha \text{-}\mathrm{amylase}\;\mathrm{inhibitory}\;\mathrm{rate}=\left(1-\frac{\left(A1-A2\right)}{\left(A3-A4\right)}\right)\times 100\%$$
where, *A*1 and *A*2 denote the absorbance values of the sample group and the sample blank group, respectively; *A*3 and *A*4 denote the absorbance values of the blank group and the blank control. Linear regression fitting was performed between the concentration of each treatment group and the corresponding inhibitory rate of α-amylase. The IC50 value was obtained.

### 2.8. Determination of Water Extracts, Total Polyphenols, Total Amino Acids, Caffeine and Soluble Sugar

The water extract content of the fruit-flavored black tea samples was determined according to the steps of the Chinese national standard, “Determination of water extract content” (GB/T 8305-2013) [35]. The 2.0 g tea powder was accurately weighed. Then, 500 mL of boiling water was added. The tea was extracted in a water bath at a constant temperature of 100 °C for 45 min. The tea broth was filtered to obtain the tea residue. The tea dregs were dried twice at 120 °C and subsequently weighed. The water extract content was 100% minus the percentage of tea dregs, which represented the weight proportion of water extract in the tea.

The total free amino acid content of fruity black tea was determined according to the method of the Chinese national standard, “Determination of free amino acid content of tea” (GB/T 8314-2013) [36]. The 3.0 g fruit-flavored black tea was accurately weighed. Then, 450 mL of boiling water was added. The extract was kept at 100 °C for 45 min in a thermostatic water bath. The extract was then filtered through three layers of 100-mesh filter cloth. The filtrate was cooled to room temperature and then volume-determined with distilled water to 500 mL. The 1 mL fruit-flavored black tea infusion, 0.5 mL phosphate buffer (pH 8.0) and 0.5 mL 2% (*w*/*v*) ninhydrin solution were prepared. The reaction was carried out in a thermostatic water bath at 100 °C for 15 min. The 25 mL distilled water was injected and equilibrated for 10 min. The absorbance value of each fruit-flavored black tea sample was analyzed at a wavelength of 570 nm. The free amino acid content of each fruit-flavored black tea was calculated from the absorbance value.

The tea polyphenol content of fruity black tea was determined according to the method of the Chinese national standard, “Determination of total polyphenol and catechin content in tea” (GB/T 23376-2018) [34]. The 0.2 g tea powder and 5 mL 70% (*v*/*v*) methanol were mixed in a constant temperature water bath at 70 °C. The 3500 RCF was performed twice for 10 min each (centrifuge 5810R, Eppendorf; Hamburg, Germany). The supernatants from the 2 centrifugations were combined. The supernatant was diluted 100-fold with distilled water. The 1 mL diluted solution was mixed and reacted with 5 mL of 10% (*v*/*v*) imine-phenol for 8 min. The 4 mL 7.5% (*w*/*v*) Na_2_CO_3_ was added and mixed well. The solution was then equilibrated for 60 min. The absorbance of individual fruity black tea broth samples was measured at a wavelength of 765 nm. Absorption spectra were measured by a UV/VIS spectrophotometer (Shimadzu UV 2550, Kyoto, Japan). A calibration curve was constructed using distilled water as the blank and gallic acid as the standard, respectively.

The content of soluble sugar in fruity black tea was determined with the anthrone sulfuric acid method. The fruity black tea sample was crushed and ground. The fruity black tea powder was accurately weighed to 100 mg and placed in a 15 mL centrifuge tube. Then, 10 mL of 80% ethanol was added and kept in the water bath at 80 °C for 30 min. The solution was centrifuged at 8000 rpm for 10 min The supernatant was transferred after concentration to a 50 mL beaker. The above extract was evaporated from ethanol on an 80 °C water bath. It was evaporated to the extent that only 3 mL of liquid remained. The liquid was fixed to 25 mL with distilled water to obtain the fruit-flavored black tea solution to be tested. Then, 5 mL of the fruity black tea infusion was put in be tested in a 25 mL volumetric flask and we fixed the volume with distilled water. Then, 2 mL of dilution solution was put into the colorimetric tube. A 10 mL anthrone reagent was slowly added and shaken well for 30 s. The reaction solution was heated in the boiling water bath for 10 min and then cooled in an ice bath box for rapid cooling for 20 min The absorbance value under the wavelength of 620 nm (OD_620_ nm) was determined in a spectrophotometer separately. The experiment was performed three times in parallel. The calibration curve y = 0.1322 x (R^2^ = 0.9998, y is concentration, x is OD_620_ nm) was used to determine the sugar amount. The content of soluble sugar in each sample was calculated according to the calibration curve.

The content of caffeine in fruit black tea was determined by liquid chromatography according to the steps of the Chinese national standard, “Determination of water extract content” (GB/T 8313-2018) [37]. The detector was a UV detector. The column was ZORBAXSB-C18 ODS (5 μm, 4.6 mm × 150 mm). Mobile phase A contained 9% acetonitrile, 2% acetic acid and 0.002% EDTA-2Na. The mobile phase B contained 80% acetonitrile, 2% acetic acid and 0.002% EDTA-2Na. The flow rate of liquid chromatography analysis was 1 mL/min and the column temperature was 35 °C. The detection wavelength was 278 nm. The injection volume was 10 μL. After 100% of phase A was kept for 10 min, the mobile phase B was changed from 0 to 32% in a linear gradient within 15 min, and then kept in the same proportion for 10 min, and finally returned to 100% of phase A. The calibration curve y = 2.46 x (R^2^ = 0.9999, y is concentration, x is peak area) had been used to determine the caffeine amount. The content of caffeine in each sample was calculated according to the calibration curve.

### 2.9. Determination of Theaflavins, Thearubigins and Theobromines of Fruit Black Tea

The theaflavins, thearubigins and theobromines were measured using a spectrophotometric method referred to in a previous study [25]. Here, 3.0 g of fruit-flavored black tea leaves were extracted in 125 mL hot water for 10 min separately. The different fruit-flavored black tea infusions were quickly cooled by an ice bath and filtered by 3 layers of filter cloth separately. Then, 30 mL of the different fruit-flavored black tea infusion was mixed with 30 mL of ethyl acetate and shaken vigorously for 3 min in the dispensing funnel separately. The mixture solution was left for 60 min after stratification. The water phase (WP) and ethyl acetate phase (EAP) were obtained respectively.

Two mL ethyl acetate phase (EAP) was determined by 95% ethanol to 25 mL, for which optical density at 380 nm was defined as E_a_.

Then, 15 mL of the different fruit-flavored black tea infusion was mixed with 15 mL of butanol in the dispensing funnel separately. The mixture solution was shaken for 5 min and then was left for 60 min after stratification separately. For the water phase, 2 mL saturated oxalic acid solution and 6 mL of water were added. Then, the mixed solution was determined by 95% ethanol to 25 mL, for which optical density at 380 nm was defined as E_b_.

Then, 15 mL of ethyl acetate phase (EAP) was mixed with 15 mL of 2.5% (*w*/*v*) NaHCO_3_ solution and shaken vigorously for 30 s in the dispensing funnel. The ethyl acetate phase solution was determined by 95% ethanol to 25 mL, for which optical density at 380 nm was defined as E_c_.

For the water phase (WP), 2 mL saturated oxalic acid solution and 6 mL water were added. Then, the mixed solution was determined by 95% ethanol to 25 mL, for which optical density at 380 nm was defined as E_d_.

The absorbance was measured at 380 nm by the UV/VIS spectrophotometer (Shimadzu UV 2550, Kyoto, Japan). A 95% ethanol solution without a test sample solution was a control treatment. The contents of theaflavins, thearubigins and theobromines were calculated by the following equation:(3)$$\mathrm{theaflavins}\%=2.25\times \frac{E_{c}}{(1-M)}$$(4)$$\mathrm{thearubigins}\%=7.06\times \frac{(2\times E_{a}+2\times E_{d}-{2E}_{b}-E_{c})}{(1-M)}$$(5)$$\mathrm{theobromines}\%=7.06\times \frac{2\times E_{b}}{\left(1-M\right)}$$

Note: M is the moisture content of tea sample.

### 2.10. Data Processing

All results were recorded as mean ± standard deviation (three replicates). The difference significances between the means were calculated by one-way analysis of variance (ANOVA) using a SPSS version 25 (SPSS Inc., Chicago, IL, USA) with a threshold of *p* < 0.05.

## 3. Results and Discussion

### 3.1. Sensory Quality Analysis of Fruit Black Tea Processed from Different Kinds of Juice

The total score of fruit black tea processed from different kinds of juice results in descending order were J5 > J4 > J3 > J2 > J1 > CK (Table 1). The total score of black tea without juice was significantly lower than that of black tea with fruit juice (*p* < 0.05), which indicated that the sensory quality of fruity black tea is superior to that of traditional black tea. The total score, tea appearance score, infusion color score, infusion taste score, infusion aroma score and tea residual score of black tea with pear juice were significantly higher than those of black tea with banana juice and apple juice (*p* < 0.05). The sensory quality of pear black tea was better than that of apple black tea and banana black tea. By comparing the sensory scores of the three kinds of pear black tea, it could be seen that the sensory quality of crown pear black tea is better than that of Sydney pear and fragrant pear. In a word, the sensory quality of crown pear black tea was the best.

From the observed experimental phenomena, compared with the control black tea without juice, the tea soup with juice was more red and bright, with a sweet fruit flavor, sweet and mellow taste and a less astringent taste. In conclusion, adding fruit juice could improve the color, taste and aroma quality of black tea soup. The color, taste and aroma scores of black tea with pear juice were higher than those with banana juice and apple juice, especially with crown pear juice.

### 3.2. Differences in Polyphenol Oxidase of Black Tea Processed with Different Juices

Comparing the polyphenol oxidase activities of different kinds of fruit juices (Figure 1), the results showed that the polyphenol oxidase activities of the fruit juices were in the order from high to low as crown pear juice > Sydney pear juice > fragrant pear juice > banana juice > apple juice. The polyphenol oxidase activities of different kinds of fruit juices significantly differed from each other (*p* < 0.05). The polyphenol oxidase activities of the crown pear juice and Sydney pear juice were significantly higher than black tea without juice (*p* < 0.05). The polyphenol oxidase activities of the fragrant pear juice were not significantly lower than black tea without juice (*p* > 0.05).

Comparison of polyphenol oxidase activity of black tea before and after fermentation with different types of fruit juices is shown in Figure 2. The polyphenol oxidase enzyme activity of fermented tea leaves without juice was significantly lower than that of fermented tea leaves with juice (*p* < 0.05). During the pre-fermentation period, the polyphenol oxidase enzyme activity of kneaded leaves with the addition of J5 was significantly higher than that of the other juice adding treatments (J1, J2, J3, J4) and CK (*p* < 0.05). During the post-fermentation period, the polyphenol oxidase enzyme activity of kneaded leaves with the addition of J5 was also significantly higher than that of the other juice adding treatments (J1, J2, J3, J4) and CK (*p* < 0.05). The polyphenol oxidase activity of the post-fermentation period was lower than that of the pre-fermentation period. Thus, the polyphenol oxidase activity of crown pear juice was high and favorable to promote the formation of black tea fermentation quality.

### 3.3. Analysis of Flavor Substances of Fruit Black Tea Processed from Different Kinds of Juice

Differences in the flavor substances of fruit black tea processed from different kinds of juice are shown in Figure 3. The free amino acid contents, water extract contents and soluble sugar contents of black tea with juices were significantly higher than those of black tea without juice (*p* < 0.05). The free amino acid contents, water extract contents and soluble sugar contents of black tea with crown pear juice were significantly higher than those of the other juice adding treatments (J1, J2, J3, J4) and CK (*p* < 0.05). The soluble sugar contents of black tea with three pear juices (J1, J2, J3) were significantly higher than those of the black tea with apple juice (J1) and banana juice (J2) (*p* < 0.05). The soluble sugar contents of black tea with Sydney pear juice (J4) were significantly higher than those of the black tea with fragrant pear juice (J3) (*p* < 0.05). There was no significant difference between the black tea with apple juice (J1) and black tea with banana juice (J2) (*p* > 0.05). The free amino acid was one of the main substances in the fresh flavor of tea infusion [18,25,30]. The water extract contents and soluble sugar contents affected the rich, mellow and sweet taste of black tea. It could be inferred that the taste of black tea with crown pear juice was more fresh, rich, mellow and sweet, which is consistent with the sensory quality evaluation of different tea tastes in Table 1.

Caffeine is one of the important sources of bitter substances in tea in the earlier study [3,19,21]. In this study, the caffeine contents of black tea with crown pear juice were not significantly similar to those of the other juice adding treatments (J1, J2, J3, J4) and CK (*p* < 0.05). There was no significant difference in the free amino acid content of J1, J2, J3, J4 (*p* > 0.05). There was no significant difference in the effect of juice addition on black tea caffeine. There was no significant difference in the effect of juice addition on black tea caffeine.

In a word, black tea with the addition of pear juice contained more water extract, soluble sugars and free amino acids than other fruit treatments. Moreover, black tea with crown pear juice had a high content of water extract, soluble sugar and free amino acids. The content of the above flavor substances increased and the flavor of black tea had been enhanced.

### 3.4. Content Analysis of Tea Polyphenols and Their Oxidized Polymers of Fruit Black Tea Processed from Different Kinds of Juice

The black tea with crown pear contained a high content of polyphenol oxidase, which could promote the enzymatic action of black tea chemicals, and thus affected the formation of its flavor quality in the earlier study [16,17,18]. In this study, differences in the tea polyphenols, theobromine, thearubigins and theaflavins of fruit black tea processed from different kinds of juice are shown in Figure 4. The thearubigins and theaflavins contents of J5 were significantly higher than those of J1, J2, J3, J4 and CK (*p* < 0.05). The tea polyphenols contents of crown pear black tea were significantly lower than traditional black tea and other fruity black tea (*p* < 0.05). The thearubigins and theaflavins content of black tea without fruit juice was significantly lower than that of black tea with juices (*p* < 0.05). The tea polyphenols, theobromine contents of black tea without fruit juice, were significantly higher than those of black tea with juices (*p* < 0.05). There was no significant difference in the theobromine contents of J1, J2, J3, J4, J5 (*p* > 0.05).

Complex enzymatic and non-enzymatic oxidations occurred during black tea processing. Tea polyphenols were oxidized and polymerized under the joint action of polyphenol oxidase, temperature and humidity to form theaflavins and thearubigins [19,20,21]. Theaflavins were a low polymerization oxidation product of tea polyphenols, while thearubigins and theobromine were high polymerization oxidation products of tea polyphenols, especially theobromine [23]. It was well known that theaflavins and thearubigins were positive for black tea flavor quality, while theobromine were negative for black tea flavor quality [8,13,15]. The black tea with crown pear contains a high content of polyphenol oxidase, which could promote the enzymatic action of black tea chemicals, and thus, affect the formation of its flavor quality. Moreover, black tea with crown pear had a high content of thearubigins and theaflavins. The higher the content of the above-mentioned tea polyphenols and their oxidized polymers, the stronger the flavor of the black tea.

### 3.5. Inhibition of α-Amylase Activity of Fruit Black Tea Processed from Different Kinds of Juice

The in vitro α-amylase inhibitory effects of different juice-added black teas were investigated. As shown in Figure 5, J5 had better α-amylase inhibitory activity. There were more studies showing that tea polyphenols and theaflavins in tea had better α-amylase inhibitory effects. Exogenous enzyme-assisted fermentation increased the activity of polyphenol oxidase, which in turn, promoted theaflavin formation and a reduction in tea polyphenols [25]. The inhibited α-amylase activity of juice-treated fruity black tea was higher than CK. The inhibition ratio of J5 was significantly higher than that of black tea without added juice treatment and that of J1, J2, J3, J4 (Figure 5a). The IC50 of J5 was as low as 3.20 U/mL ± 0.02. The IC50 value of J5 was significantly higher than that of black tea without added juice treatment and that of juice-treated J1, J2, J3, J4 (Figure 5b).

The changing pattern of the α-amylase inhibitory rate of black tea with different fruit juice treatments was consistent with the changing trends of tea polyphenols, soluble sugars, theaflavins, thearubigins and theaflavins. More studies had shown that tea polyphenols, soluble sugars, and theaflavins have inhibitory effects on α-amylase. Among the 36 differential compounds labeled, most of the tea polyphenols, catechins, dimeric catechins, and flavonoid glycosides were reported to have a hypoglycemic effect [32]. The results of this experiment showed that the fruit-flavored black tea treated with crown pear juice contributed to the depletion of tea polyphenols and the formation of theaflavins and thearubigins, resulting in a better hypoglycemic effect of the fruit-flavored black tea.

## 4. Conclusions

Fruit-flavored black tea processing is an important way to solve the problems of heavy bitterness and astringency, poor flavor quality and the low functional activity of summer and autumn tea. The pear, apple and banana were easily obtained and overabundant. In order to improve the bitter and astringent taste of black tea in summer and autumn, the pear, apple and banana were selected and added to the black tea fermentation process. The flavor quality and sugar reduction function of black tea through the intervention of fruit juices enriched with polyphenol oxidase improved the fresh and sweet flavor. We hope to provide a theoretical basis for the regulation of black tea quality and the research and development of functional black tea beverages.

These results showed that fruit-flavored black teas were more red, sweet and mellow. It was surprising that bitterness and astringency decreased significantly. The sensory quality evaluation scores of fruit-flavored black teas were significantly higher than those of black tea treated without fruit juice, especially those of fruit-flavored black tea treated with crown pear juice. The polyphenol oxidase activity, water extract, soluble sugar, free amino acid, theaflavin, thearubigin and inhibition of amylase activity of fruit-flavored black tea were significantly higher than those of black tea without fruit juice (*p* < 0.05). The contents of tea polyphenols in the fruit-flavored black tea were significantly lower than those of the black tea treated without fruit juice (*p* < 0.05). The sensory evaluation scores, polyphenol oxidase activity, water extract, soluble sugar, free amino acid, theaflavin, thearubigin and inhibition of amylase activity of the fruit-flavored black tea treated with pear juice were significantly higher than those of the fruit-flavored black tea treated with apple and banana juices (*p* < 0.05), especially crown pear juice. The relatively inexpensive apples and pears were introduced into black tea processing. The principles of fruit–tea composite fermentation and fruit–tea flavor fusion were applied to effectively reduce the bitterness and astringency of black tea in summer and autumn. The flavor was significantly improved and its hypoglycemic activity was increased. The research hoped to provide a theoretical basis for black tea quality control research, functional black tea beverage research and development, and summer and autumn tea resource utilization technology research.

## Figures and Tables

**Figure 1 foods-14-00588-f001:**
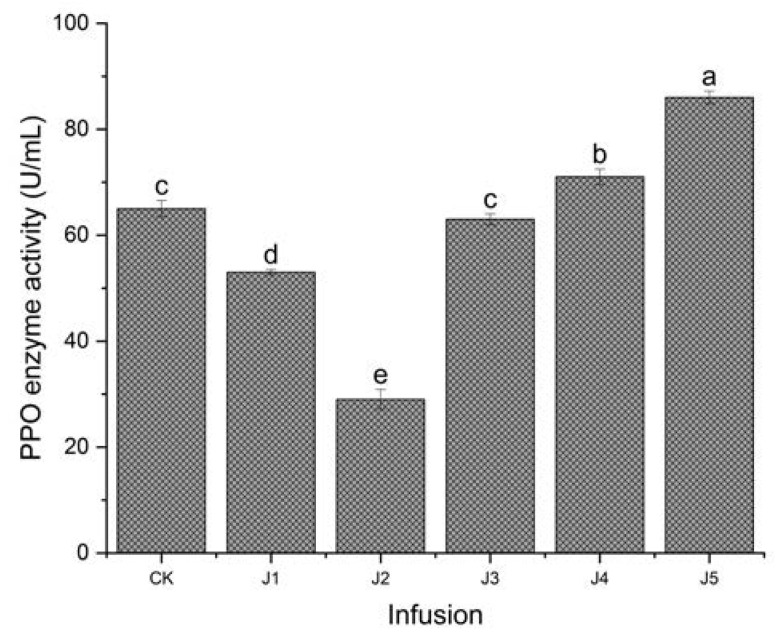
Comparison of polyphenol oxidase activities of different types of fruit juices (Note: CK refers to black tea without juice treatment, J1 refers to black tea treated with banana juice, J2 refers to black tea with apple juice treatment, J3 refers to black tea with fragrant pear juice, J4 refers to black tea with Sydney pear juice, J5 refers to black tea with crown pear juice. The error bars indicate the standard deviation of the mean for *n* = 3 samples, and lowercase letters represent significant differences between samples in each column (*p* < 0.05)).

**Figure 2 foods-14-00588-f002:**
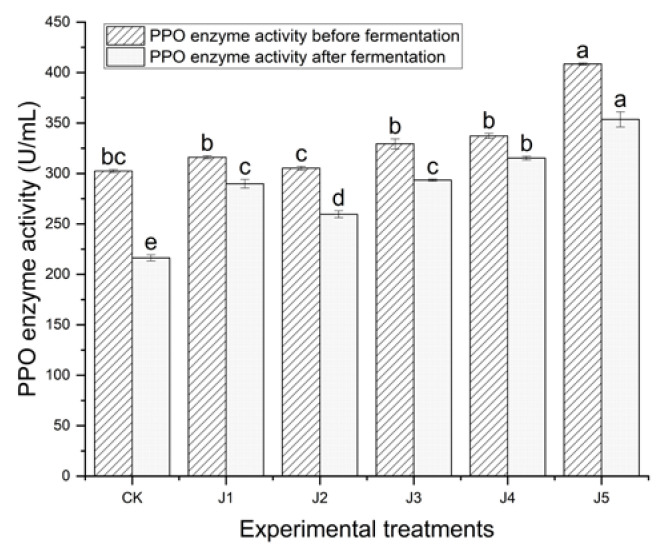
Comparison of polyphenol oxidase activities of black tea before and after pre-fermentation with the addition of different kinds of fruit juices (note: CK refers to black tea without juice treatment, J1 refers to black tea treated with banana juice, J2 refers to black tea with apple juice treatment, J3 refers to black tea with fragrant pear juice, J4 refers to black tea with Sydney pear juice, J5 refers to black tea with crown pear juice. The error bars indicate the standard deviation of the mean for *n* = 3 samples, and lowercase letters represent significant differences between samples in each column (*p* < 0.05)).

**Figure 3 foods-14-00588-f003:**
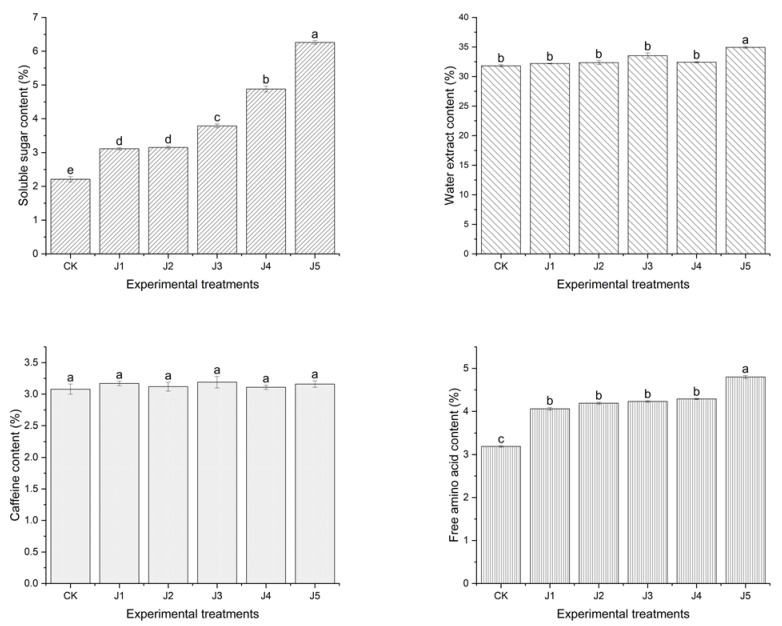
Free amino acid, water extract, soluble sugar and caffeine contents of fruit-flavored black tea fermented with different kinds of fruit juices (note: CK refers to black tea without juice treatment, J1 refers to black tea treated with banana juice, J2 refers to black tea with apple juice treatment, J3 refers to black tea with fragrant pear juice, J4 refers to black tea with Sydney pear juice, J5 refers to black tea with crown pear juice. The error bars indicate the standard deviation of the mean for *n* = 3 samples, and lowercase letters represent significant differences between samples in each column (*p* < 0.05)).

**Figure 4 foods-14-00588-f004:**
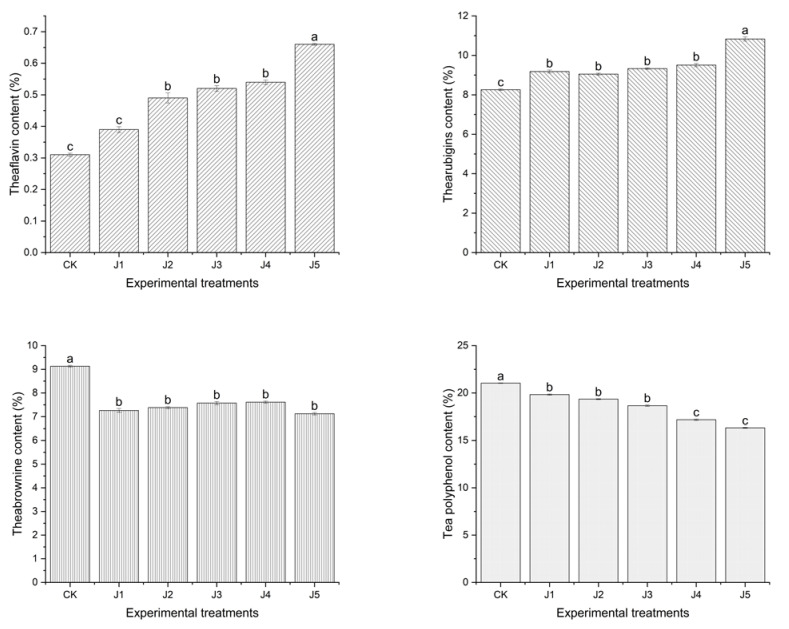
Tea polyphenols and their oxidized polymers in fruit-flavored black teas fermented with different kinds of fruit juices (note: CK refers to black tea without juice treatment, J1 refers to black tea treated with banana juice, J2 refers to black tea with apple juice treatment, J3 refers to black tea with fragrant pear juice, J4 refers to black tea with Sydney pear juice, J5 refers to black tea with crown pear juice. The error bars indicate the standard deviation of the mean for *n* = 3 samples, and lowercase letters represent significant differences between samples in each column (*p* < 0.05)).

**Figure 5 foods-14-00588-f005:**
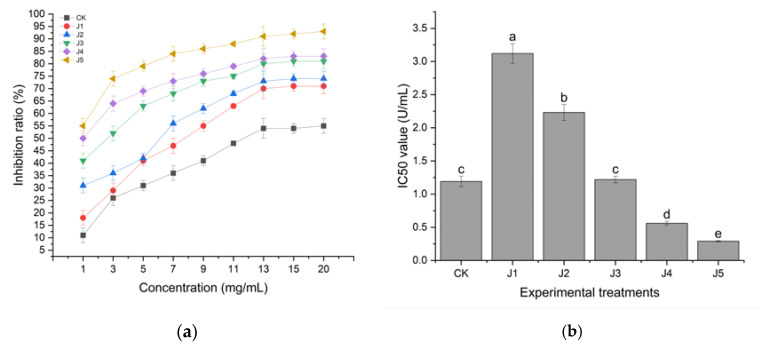
Inhibitory rate (**a**) and IC50 value (**b**) of α-amylase activity of fruit black tea processed from different kinds of juice (note: CK refers to black tea without juice treatment, J1 refers to black tea treated with banana juice, J2 refers to black tea with apple juice treatment, J3 refers to black tea with fragrant pear juice, J4 refers to black tea with Sydney pear juice, J5 refers to black tea with crown pear juice. The error bars indicate the standard deviation of the mean for *n* = 3 samples, and lowercase letters represent significant differences between samples in each column (*p* < 0.05)).

**Table 1 foods-14-00588-t001:** Sensory quality of black tea with different fruit juice.

Sample	Tea Appearance		Tea Infusion Color		Tea Infusion Aroma		Tea Infusion		Tea Residual		Total Score
	Description	Score (10%)	Description	Score (20%)	Description	Score (35%)	Description	Score (35%)	Description	Score (10%)	
Black tea without juice (CK)	tight knot and slightly gold	87.0 ± 0.1 ^c^	orange-red and brighter	90.2 ± 0.1 ^d^	highly fresh and slightly astringent	88.0 ± 0.7 ^d^	rich and mellow	88.1 ± 0.4 ^d^	Red, soft, bright, more uniform	89.1 ± 0.4 ^d^	88.1 ± 0.7 ^d^
Black tea with banana juice (J1)	tight knot and slightly gold	88.0 ± 0.2 ^b^	orange-red and brighter	92.1 ± 0.8 ^c^	highly fresh and slightly sweet	88.1 ± 0.4 ^e^	fresh and mellow	89.3 ± 0.3 ^c^	red, soft, bright and still well-balanced.	90.3 ± 0.3 ^c^	89.0 ± 0.4 ^c^
Black tea with apple juice (J2)	fine, even emerald green	89.2 ± 0.1 ^a^	intense green	93.0 ± 0.5 ^b^	highly fresh, slightly fruity	89.1 ± 0.2 ^d^	fresh and mellow, slightly sweet	89.2 ± 0.6 ^c^	red, soft, bright and even	91.2 ± 0.6 ^b^	89.3 ± 0.2 ^c^
Black tea with fragrant pear juice (J3)	fine, even emerald green	88.1 ± 0.3 ^b^	intense green	92.2 ± 0.2 ^c^	highly fresh, slightly fruity	91.1 ± 0.3 ^b^	fresh and mellow, slightly sweet	90.4 ± 0.5 ^b^	red, soft, bright and even	92.3 ± 0.5 ^a^	90.1 ± 0.5 ^b^
Black tea with Sydney pear juice (J4)	fine, even emerald green	89.1 ± 0.4 ^a^	intense green	93.2 ± 0.6 ^b^	highly fresh, fruity	90.1 ± 0.5 ^c^	fresh and mellow, sweet	90.1 ± 0.7 ^b^	red, soft, bright and even	92.4 ± 0.7 ^a^	90.3 ± 0.6 ^b^
Black tea with crown pear juice (J5)	tight knot and slightly gold	89.2 ± 0.2 ^a^	orange-red and brighter	94.3 ± 0.6 ^a^	highly fresh and more sweet	92.0 ± 0.5 ^a^	more fresh, rich and mellow, more sweet	92.4 ± 0.7 ^a^	red, soft, bright and still well-balanced	92.6 ± 0.7 ^a^	91.5 ± 0.6 ^a^

Note: lowercase letters represent significant differences between samples in each column (*p* < 0.05).

## Data Availability

The original contributions presented in the study are included in the article; further inquiries can be directed to the corresponding author.

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
