# Peer review of "Effects of Different Kinds of Fruit Juice on Flavor Quality and Hypoglycemic Activity of Black Tea"

_foods, 2025, doi:10.3390/foods14040588_

Round 1

Reviewer 1 Report

Comments and Suggestions for Authors

Abstract

Try to divide long periods into shorter sentences to improve readability. Review the

use of “P<0.01” in italics, in accordance with scientific norms. I suggest mentioning in the

summary the fruits used to make the juices: bananas, apples, fragrant pear and sydney pear.

Materials and methods

At “2.3. Preparation of fresh fruit juice containing polyphenol oxidase (PPO)” make it

clear whether it is a mixture with all the fruits or a mixture for each fruit. Make it clear also in

section “2.5. Processing of Fruity Black Tea”.

I suggest considering changing the terms “sorbets” and “crown pears (line 125) to

“fragrant pear” and “sydney pear”, respectively, as they appear in the Results.

I suggest changing the sub-title “2.5. Processing of Fruity Black Tea” (line 136) to

lowercase as standard.

In this same item, revise the text by putting the verbs in the past tense, especially

those found between lines 136 and 142. Complete the equipment information by adding the

city.

Make sure that the term “LTD” is written correctly throughout the article.

In section “2.4. Sensory evaluation of fruity black tea” should mention more clearly

how the determination was made for the qualitative and quantitative assessment of the

quality characteristics of the fruity black tea. And what is the “code review method”?

The information “The fruity black tea was assigned 25% for tea appearance, 10% for

tea infusion color, 25% for infusion aroma, 30% for tea infusion taste and 10% for tea residue.’

(lines 158/159) should be in the Results section.

The black tea samples (line 159) were also prepared according to the China National

Standard?

Why did you work with 7 professional tea tasters for fruity black tea and 5 professional

tea tasters for the black tea? And why were they differentiated between male and female just

for the 5´s?

Line 162, the liquor color, aroma and taste were analyzed in what way?

“Formula 1” was mentioned in line 164 but the formula is not named in line 165. I also

suggest changing the term “formula” to “equation”. Add at the beginning of line 180 “Where,”.

Quote equation 2 on line 178 and name it on line 179.

Please change the centrifugal rotation unit to the international standard RCF (Relative

Centrifugal Force) (line 209).

Certify at line 216 if the sentence was correct, or should be something like “A

calibration curve was constructed using distilled water as the blank and gallic acid as the

standard, respectively.”.

Verify the term “min.” remove the dot from the abbreviation for minutes throughout

the text.

Line 233, add the information from ZORBAXSB-C18 column.

Review the sentence “The separation was performed in a gradient of 0 to 32% for 15

min, and then in a gradient of 0 to 32% for 15 min.” (lines 240/241).

Add “and” between “thearubigins, theobromines” (line 243 and 262).

The abbreviations TF, TR and TB (line 244) were not used throughout the text, so I

suggest leaving them in full or using the abbreviations later.

Line 245, are you sure you meant “wastewate” in this sentence?

In section “2.7. Determination of theaflavins, thearubigins, theobromines of fruit black

tea” check what has been named as Ec (lines 254 and 256), Ed (line 256), Eb (line 260) and

check with the equations mentioned. Verify the unnecessary dots (.) on lines 249, 253, 245,

255, 256, and throughout the text.

I think something is missing between “Ec. mL” (line 256).

Line 262, take off “a” from “a blank”.

Line 263, “he following equation” Cite the equations numbers and named each one.

Data processing

Line 240, I suggest add (ANOVA) and the significance level.

Results

Section “3.1. Sensory quality analysis of fruit black tea processed from different kinds

of juice”:

Pay attention to the abbreviations (J5, J4, J3, J2, J1 and CK) you use. In the text, name

the abbreviation at the beginning and then use only the abbreviation.

How can you affirm this “the tea soup with juice is more red and bright, with sweet

fruit flavor, sweet and mellow taste, and less astringent taste.” In lines 277/278 if the Table 1

does not show this information?

In fact, the “tea appearance score” from of black tea with pear juice J3 had no

significant difference whit black tea with banana juice. Also the information on lines 281 and

282 “The total score, infusion color score, infusion taste score, infusion aroma score and tea

residual score of black tea with pear juice was significantly higher than that of black tea with

apple juice” should be checked carefully.

Line 287, “with sydney pear juice” and with crown pear juice.

Line 289, … and Tea residual score.

Where is the data for describe the reported at lines 292 to 316 for the flavor quality

characteristics of black tea with and without juice?

Check the information on lines 318 and 319, the statistically greater was only the tea

with crown pear juice.

I suggest standardizing the citation of treatments in the text according to the sequence

shown in the table.

Section “3.2. Differences in polyphenol oxidase of black tea processed with different

juices”

Line 327, “are” should be “were”.

Lines 328 and 330, I think that “tea juice” could be “black tea without juice”, please,

check.

At Figure 1, you don't need a “note” if you haven't used any abbreviations in the

figure. But I suggest putting the abbreviations in the figure, since you used abbreviations in

figure 2.

Section “3.3. Analysis of flavor substances of fruit black tea processed from different

kinds of juice”

The mentioned figure in the text was not the same, should be Figure 3, please check

and put in order.

Section “3.4. Content Analysis of tea polyphenols and their oxidized polymers of fruit

black tea processed from different kinds of juice”

The mentioned figure in the text was not the same, should be Figure 4, please check

and put in order.

Line 387, “thearubigins and theaflavin” not “thearubigins, theaflavin”.

With regard to statistics, sometimes the letter a is used for the highest value or

sometimes for the lowest.

Line 393, “black tea” is repeated.

Line 404, check the font size.

Check is information “Addition of fruit juice during black tea processing increased the

theaflavin content, theobromine content and decreased theaflavin content of black tea

compared with the control.” (lines 404-406) because you can only say that the addition of fruit

juices has increased if statistically there has been this increase, and this has only happened for

thearubigins.

Line 407, is not higher.

The information on line 409 was already in line 387.

Section “3.5. Inhibition of α-amylase activity of of fruit black tea processed from

different kinds of juice”

Sub-title if two “of”.

I suggest removing the first sentence because it only takes up part of the objectives of

the work and this has not been taken up in all the previous topics.

Line 420, standardize the text to “Fig.” or “Figure”. Check all.

Now you used the term “fruit-flavored black tea” (line 420, 436, 445, 456 and 457) or

“juice-treated fruity black tea” (line 425). Also started to use “crown pear-treated fruity black

tea” (lines 426, 427 and 428).

Line 421 and 422, “α amylase” without “-”.

I suggest checking the information “In this experiment, the juice-treated fruity black

tea inhibited α-amylase activity higher than that of black tea without added juice treatment,

especially the crown pear-treated fruity black tea.” (lines 425-427). The figure 5a does not

show two curves (from CK and J5) in the graph.

Are you sure that it was the treatment with crown pear fruit, not the banana fruit? In

this phrase, “The IC50 of crown pear-treated fruity black tea was as high as 32 U/mL ± 40.25.

The IC50 of crown pear-treated fruity black tea was significantly higher than that of black tea

without added juice treatment and that of juice-treated J1, J2, J3, J4.” (lines 427-429).

About statistics, sometimes the letter “a” was used for the highest value and

sometimes for the lowest. Standardize it.

Conclusions

“In this study” was used to many times.

Line 459, should be “α-amylase” not “amylase”.

I suggest revising the conclusion to improve the flow and connectivity between

sentences, as they currently feel disjointed and repetitive.

Comments on the Quality of English Language

The article must be revised to improve the quality of the English language.

Author Response

Author's Reply to the Review Comment of Reviewer 1

Comments 1:Try to divide long periods into shorter sentences to improve readability. Review the use of “P<0.01” in italics, in accordance with scientific norms. I suggest mentioning in the summary the fruits used to make the juices: bananas, apples, fragrant pear and sydney pear. 

Response 1:Thanks for your good suggestions. The long sentences of the manuscript have been converted into short sentences. The abstract and the text of the P letter have been revised in italics. The names of different fruits have also been added to the summary. Red font parts were the result of modified.

Comments 2:At “2.3. Preparation of fresh fruit juice containing polyphenol oxidase (PPO)” make it clear whether it is a mixture with all the fruits or a mixture for each fruit. Make it clear also in section “2.5. Processing of Fruity Black Tea”.

Response 2:Thanks for your good suggestions. It is a mixture for each fruit. According to your suggestion,  “2.3. Preparation of fresh fruit juice containing polyphenol oxidase (PPO)” and  “2.5. Processing of Fruity Black Tea” have been revised. Red font parts were the result of modified. Red font parts were the result of modified.

Comments 3:I suggest considering changing the terms “sorbets” and “crown pears (line 125) to "fragrant pear" and “sydney pear”, respectively, as they appear in the Results.

Response 3:Thanks for your good suggestions. The terms “bananas, apples, pear, sorbets and crown pears” (line 125) had been changed to "banana, apple, fragrant pear, sydney pear and crown pear" . Red font parts were the result of modified.

Comments 4:I suggest changing the sub-title “2.5. Processing of Fruity Black Tea” (line 136) to lowercase as standard. 

Response 4:Thanks for your good suggestions. According to your suggestion, the sub-title “2.5. Processing of Fruity Black Tea” (line 136) had been revised for lowercase. Red font parts were the result of modified. 

Comments 5:In this same item, revise the text by putting the verbs in the past tense, especially those found between lines 136 and 142. Complete the equipment information by adding the city.

Response 5:Thanks for your good suggestions. According to your suggestion, the verbs had been revised in the past tense. The city had been added to the equipment information. Red font parts were the result of modified.

Comments 6:Make sure that the term “LTD” is written correctly throughout the article.

Response 6:Thanks for your good suggestions. According to your suggestion, the term “LTD” had been written correctly throughout the article..

Comments 7:In section “2.4. Sensory evaluation of fruity black tea” should mention more clearly

how the determination was made for the qualitative and quantitative assessment of the

quality characteristics of the fruity black tea. And what is the “code review method”?

Response 7:Thanks for your good suggestions. “code review method” should be crypto sensory evaluation method, which had been revised in manuscript. Red font parts were the result of modified.

Comments 8:The information “The fruity black tea was assigned 25% for tea appearance, 10% for tea infusion color, 25% for infusion aroma, 30% for tea infusion taste and 10% for tea residue.’ (lines 158/159) should be in the Results section.

Response 8:Thanks for your good suggestions. According to the China National Standard (GB/T 23776–2018), the fruity black tea was assigned 25% for tea appearance, 10% for tea infusion color, 25% for infusion aroma, 30% for tea infusion taste and 10% for tea residue. These were not the results. These just the evaluation method of tea. Red font parts were the result of modified.

Comments 9:The black tea samples (line 159) were also prepared according to the China National Standard?

Response 9:Thanks for your good questions. According to the China National Standard (GB/T 23776–2018), the flavor quality of black teas were evaluation. According to your suggestion, the sentence (line 159) have been revised. Red font parts were the result of modified.

Comments 10:Why did you work with 7 professional tea tasters for fruity black tea and 5 professional tea tasters for the black tea? And why were they differentiated between male and female just for the 5´s?

Response 10:Thanks for your good questions. Both traditional black tea and fruity black tea were evaluated by 5 professional tea tasters. 7 professional tea tasters was a clerical error. It has been corrected. Red font parts were the result of modified.

Comments 11:Line 162, the liquor color, aroma and taste were analyzed in what way?

Response 11:Thanks for your good questions. The liquor color was evaluated by human professional tea tasters eyes. The aroma and taste were evaluated by human professional tea tasters tongue and nose.

Comments 12:“Formula 1” was mentioned on line 164 but the formula is not named on line 165. I also suggest changing the term “formula” to “equation”. Add at the beginning of line 180 “Where,”.

Response 12:Thanks for your good suggestions. According to your suggestion, the term “formula” had been revised to “equation”.  “Where,” had been add at the beginning of line 180. Red font parts were the result of modified.

Comments 13:Quote equation 2 on line 178 and name it on line 179.

Response 13:Thanks for your good suggestions. According to your suggestion, Quote equation 2 had been quoted on line 178 and named it on line 179. Red font parts were the result of modified.

Comments 14:Please change the centrifugal rotation unit to the international standard RCF (Relative Centrifugal Force) (line 209).

Response 14:Thanks for your good suggestions. According to your suggestion, the the centrifugal rotation unit had been revised for the international standard RCF (Relative Centrifugal Force). Red font parts were the result of modified.

Comments 15:Certify at line 216 if the sentence was correct, or should be something like “A calibration curve was constructed using distilled water as the blank and gallic acid as the standard, respectively.”

Response 15:Thanks for your good suggestions. According to your suggestion, the sentence had been changed for “A calibration curve was constructed using distilled water as the blank and gallic acid as the standard, respectively.” Red font parts were the result of modified.

Comments 16:Verify the term “min.” remove the dot from the abbreviation for minutes throughout the text.

Response 16:Thanks for your good suggestions. According to your suggestion, the dot from the abbreviation for minutes throughout the text had been revised. Red font parts were the result of modified.

Comments 17:Line 233, add the information from ZORBAXSB-C18 column.

Response 17:Thanks for your good suggestions. According to your suggestion, the information from ZORBAXSB-C18 column had been added. Red font parts were the result of modified.

Comments 18:Review the sentence “The separation was performed in a gradient of 0 to 32% for 15 min, and then in a gradient of 0 to 32% for 15 min.” (lines 240/241).

Response 18:Thanks for your good suggestions. The sentence “The separation was performed in a gradient of 0 to 32% for 15 min, and then in a gradient of 0 to 32% for 15 min.” had been deleted.

Comments 19:Add “and” between “thearubigins, theobromines” (line 243 and 262).

Response 19:Thanks for your good suggestions. According to your suggestion, “and” had been added between “thearubigins, theobromines” . Red font parts were the result of modified.

Comments 20:The abbreviations TF, TR and TB (line 244) were not used throughout the text, so I suggest leaving them in full or using the abbreviations later.

Response 20:Thanks for your good suggestions. According to your suggestion, the abbreviations TF, TR and TB had been revised. Red font parts were the result of modified.

Comments 21:Line 245, are you sure you meant “wastewate” in this sentence?

Response 21:Thanks for your good suggestions. According to your suggestion,  “wastewate” in this sentence had been deleted.

Comments 22:In section “2.7. Determination of theaflavins, thearubigins, theobromines of fruit black tea” check what has been named as Ec (lines 254 and 256), Ed (line 256), Eb (line 260) and check with the equations mentioned. Verify the unnecessary dots (.) on lines 249, 253, 245, 255, 256, and throughout the text.

Response 22:Thanks for your good suggestions. According to your suggestion, the Ea, Eb, Ec, Ed had been checked. The paragraph had been rewritten. The unnecessary dots (.) on lines 249, 253, 245, 255, 256, and throughout the text had been verified. Red font parts were the result of modified. Red font parts were the result of modified.

Comments 23:I think something is missing between “Ec. mL” (line 256).

Response 23:Thanks for your good suggestions. According to your suggestion, “Ec. mL” have been revised in detail. The paragraph had been rewritten. Red font parts were the result of modified.

Comments 24:Line 262, take off “a” from “a blank”.

Response 24:Thanks for your good suggestions. According to your suggestion, “a” from “a blank” had been taken off .

Comments 25:Line 263, “he following equation” Cite the equations numbers and named each one.

Response 25:Thanks for your good suggestions. According to your suggestion, the “the following equation”  had been cited the equations numbers and named each one. Red font parts were the result of modified.

Comments 26:Line 240, I suggest add (ANOVA) and the significance level.

Response 26:Thanks for your good suggestions. According to your suggestion, the (ANOVA) and the significance level had been added. Red font parts were the result of modified.

Comments 27:Section “3.1. Sensory quality analysis of fruit black tea processed from different kinds of juice”: Pay attention to the abbreviations (J5, J4, J3, J2, J1 and CK) you use. In the text, name the abbreviation at the beginning and then use only the abbreviation.

Response 27:Thanks for your good suggestions. According to your suggestion, the abbreviation was used only the abbreviation. Red font parts were the result of modified.

Comments 28:How can you affirm this “the tea soup with juice is more red and bright, with sweet fruit flavor, sweet and mellow taste, and less astringent taste.” on lines 277/278 if the Table 1 does not show this information? In fact, the “tea appearance score” from of black tea with pear juice J3 had no significant difference whit black tea with banana juice. Also the information on lines 281 and 282 “The total score, infusion color score, infusion taste score, infusion aroma score and tea residual score of black tea with pear juice was significantly higher than that of black tea with apple juice” should be checked carefully.

Response 28:Thanks for your good suggestions. According to the terminology and ratings of the five reviewers for the sensory qualities of the tea broth, we conclude that “the tea soup with juice is more red and bright, with sweet fruit flavor, sweet and mellow taste, and less astringent taste.” We have put the terms and scores of the sensory quality evaluation of tea broth into Table 1. It looks clearer. Red font parts were the result of modified.

Comments 29:Line 287, “with sydney pear juice” and with crown pear juice.

Response 29:Thanks for your good suggestions. According to your suggestion, the sentence on line 287 have been rewritten in detail and comprehensively. Red font parts were the result of modified.

Comments 30:Line 289, … and Tea residual score. Where is the data for describe the reported at lines 292 to 316 for the flavor quality characteristics of black tea with and without juice?

Response 30:Thanks for your good suggestions. The data for describe the reported at lines 292 to 316 for the flavor quality characteristics of black tea with and without juice could be seen in new table 1. Sensory evaluation of black tea infusion has been streamlined and condensed. Red font parts were the result of modified.

Comments 31:Check the information on lines 318 and 319, the statistically greater was only the tea with crown pear juice. I suggest standardizing the citation of treatments in the text according to the sequence shown in the table.

Response 31:Thanks for your good suggestions. According to your suggestion, the citation of treatments in the text according to the sequence shown in the table 1 have been revised in detail and comprehensively. Red font parts were the result of modified.

Comments 32:Section “3.2. Differences in polyphenol oxidase of black tea processed with different juices”. Line 327, “are” should be “were”. Lines 328 and 330, I think that “tea juice” could be “black tea without juice”, please, check.

Response 32:Thanks for your good suggestions. According to your suggestion, the “are” had been changed to  “were”. “tea juice” had been changed to “black tea without juice”. Red font parts were the result of modified.

Comments 33:At Figure 1, you don't need a “note” if you haven't used any abbreviations in the figure. But I suggest putting the abbreviations in the figure, since you used abbreviations in figure 2.

Response 33:Thanks for your good suggestions. According to your suggestion, the abbreviations in the figure had been used. Red font parts were the result of modified.

Comments 34:Section “3.3. Analysis of flavor substances of fruit black tea processed from different kinds of juice”. The mentioned figure in the text was not the same, should be Figure 3, please check and put in order.

Response 34:Thanks for your good suggestions. According to your suggestion, the mentioned figure in the text was Figure 3, which had been revised. Red font parts were the result of modified.

Comments 35:Section “3.4. Content Analysis of tea polyphenols and their oxidized polymers of fruit black tea processed from different kinds of juice”. The mentioned figure in the text was not the same, should be Figure 4, please check and put in order.

Response 35:Thanks for your good suggestions. According to your suggestion, the mentioned figure in the text was Figure 4, which had been revised. Red font parts were the result of modified.

Comments 36:Line 387, “thearubigins and theaflavin” not “thearubigins, theaflavin”. With regard to statistics, sometimes the letter a is used for the highest value or sometimes for the lowest.

Response 36:Thanks for your good suggestions. According to your suggestion, the “thearubigins, theaflavin” have been revised for “thearubigins and theaflavin”. The full manuscript had been changed to the letter a for the highest value. Red font parts were the result of modified.

Comments 37:Line 393, “black tea” is repeated. Line 404, check the font size.

Response 37:Thanks for your good suggestions. According to your suggestion, the sentence on line 393 have been revised. The the font size in 404 had been checked. Red font parts were the result of modified.

Comments 38:Check is information “Addition of fruit juice during black tea processing increased the theaflavin content, theobromine content and decreased theaflavin content of black tea compared with the control.” (lines 404-406) because you can only say that the addition of fruit juices has increased if statistically there has been this increase, and this has only happened for thearubigins.

Response 38:Thanks for your good suggestions. According to your suggestion, the sentence had been revised for “The thearubigins, theaflavin contents of J5 was significantly higher than that of J1, J2, J3, J4 and CK (P<0.05)”. Red font parts were the result of modified.

Comments 39:Line 407, is not higher. The information on line 409 was already on line 387.

Response 39:Thanks for your good suggestions. According to your suggestion, the information on line 409 had been deleted. Red font parts were the result of modified.

Comments 40:Section “3.5. Inhibition of α-amylase activity of of fruit black tea processed from different kinds of juice” Sub-title if two “of”. I suggest removing the first sentence because it only takes up part of the objectives of the work and this has not been taken up in all the previous topics.

Response 40:Thanks for your good suggestions. According to your suggestion, the first “of” in sentence had been deleted.  it only takes up part of the objectives of the work and this has not been taken up in all the previous topics. Red font parts were the result of modified.

Comments 41:Line 420, standardize the text to “Fig.” or “Figure”. Check all.

Response 41:Thanks for your good suggestions. According to your suggestion, the “Fig.” had been used in standardize. Red font parts were the result of modified.

Comments 42:Now you used the term “fruit-flavored black tea” (line 420, 436, 445, 456 and 457) or “juice-treated fruity black tea” (line 425). Also started to use “crown pear-treated fruity black tea” (lines 426, 427 and 428).

Response 42:Thanks for your good suggestions. According to your suggestion, the “fruit-flavored black tea” had been used revised in standardize. “crown pear-treated fruity black tea” had been changed for J5. Red font parts were the result of modified.

Comments 43:Line 421 and 422, “α amylase” without “-”. I suggest checking the information “In this experiment, the juice-treated fruity black tea inhibited α-amylase activity higher than that of black tea without added juice treatment, especially the crown pear-treated fruity black tea.” (lines 425-427).

Response 43:Thanks for your good suggestions. According to your suggestion, the “α- amylase” had been used in standardize. The sentence “In this experiment, the juice-treated fruity black tea inhibited α-amylase activity higher than that of black tea without added juice treatment, especially the crown pear-treated fruity black tea.” had been revised. Red font parts were the result of modified.

Comments 44:The figure 5a does not show two curves (from CK and J5) in the graph. Are you sure that it was the treatment with crown pear fruit, not the banana fruit? In this phrase, “The IC50 of crown pear-treated fruity black tea was as high as 32 U/mL ± 40.25. The IC50 of crown pear-treated fruity black tea was significantly higher than that of black tea without added juice treatment and that of juice-treated J1, J2, J3, J4.” (lines 427-429). About statistics, sometimes the letter “a” was used for the highest value and sometimes for the lowest. Standardize it.

Response 44:Thanks for your good suggestions. The vertical coordinate of Fig. 5a was the inhibition rate, not the IC50 value. The vertical coordinate of Fig. 5b was the IC50 value. Red font parts were the result of modified. The inhibition rate of treatment with crown pear fruit was highest. The letter “a” was used for the highest value, which could be seen in revised Fig. 5b.

Comments 45:“In this study” was used to many times.

Response 45:Thanks for your good suggestions. According to your suggestion, the “In this study” have been deleted in manuscript. Red font parts were the result of modified.

Comments 46:Line 459, should be “α-amylase” not “amylase”.

Response 46:Thanks for your good suggestions. According to your suggestion, the “amylase” had been changed for “α-amylase” . Red font parts were the result of modified.

Comments 47:I suggest revising the conclusion to improve the flow and connectivity between sentences, as they currently feel disjointed and repetitive.

Response 47:Thanks for your good suggestions. According to your suggestion, the conclusion had been revised to improve the flow and connectivity. Red font parts were the result of modified.

Comments 48:The article must be revised to improve the quality of the English language.

Response 48:Thanks for your good suggestions. According to your suggestion, English vocabulary and grammar had been changed throughout. Red font parts were the result of modified.

Reviewer 2 Report

Comments and Suggestions for Authors

This is an interesting work, but with a very poor discussion and no clear indication of scientific novelty

I suggest expanding the discussion (in this form only 3 publications discussed and that's anyway  in a very limited way)

I also suggest to clearly indicate what is an original achievement of the presented study

Additional details:

Line 117 “Clean bananas, apples, pears, sorbets and crown pears were peeled and homogenized” - What kind of sorbets? How was it peeled?

all varieties of fruit used in the experiment should be listed in the methods part

line 136: Then, the leaves were placed in a thermostat (Shanghai Yuming Instrument Co., LTD, China) at 35°C for fermentation – how long was the fermentation process conducted?

Line 137 Finally, the tea leaves were thoroughly dried - please present some parameters of drying: like temperature, time, …..

Line 236 ………….method referred to a previous study – references needed

Line 237 method referred to a previous study – what kind of wastewater?

Point 2.7 line 234-255 - one form of presentation should be maintained: “was added….”, “was cooled…..” and then: “take 15 mL of the ethyl 242 acetate layer and mix it with …….” – choose one form proper for the journal and be consistent and start a new sentence with a capital letter.

Author Response

Comments 1: I suggest expanding the discussion (in this form only 3 publications discussed and that's anyway in a very limited way). I also suggest to clearly indicate what is an original achievement of the presented study.

Response 1: Thanks for your good suggestions. According to your suggestion, we have expanding the discussion of manuscript. The original achievement of this study have been clearly revised. Red font parts were the result of modified.

Comments 2: Line 117 “Clean bananas, apples, pears, sorbets and crown pears were peeled and homogenized” - What kind of sorbets? How was it peeled?

Response 2: Thank you for pointing out the error. “sorbets” just is a wrong word. “sorbets” had been revised for “sydney pear” .  Red font parts were the result of modified.

Comments 3: all varieties of fruit used in the experiment should be listed in the methods part.

Response 3: Thanks for your good suggestions. According to your suggestion, all varieties of fruit used in the experiment have been listed in the methods part, which could be seen in line 113, 114, 115, 124, 151, 152, 153. Red font parts were the result of modified.

Comments 4: line 136: Then, the leaves were placed in a thermostat (Shanghai Yuming Instrument Co., LTD, China) at 35°C for fermentation – how long was the fermentation process conducted?  

Response 4: Thanks for your good suggestions. The fermentation process times was 6 h. According to your suggestion, this has been revised in the manuscript. The tea leaves were fermented by a thermostat (Shanghai Yuming Instrument Co., LTD, China) at 35 °C for 6 h.  Red font parts were the result of modified.

Comments 5: Line 137 Finally, the tea leaves were thoroughly dried - please present some parameters of drying: like temperature, time, …..

Response 5: Thanks for your good suggestions. The parameters of initial drying treatment 105-110 °C temperature 20 min. The parameter of redrying treatment is redrying at 80-90 °C for 60min. According to your suggestion, the parameters of drying have been added. Red font parts were the result of modified.

Comments 6: Line 236 ………….method referred to a previous study – references needed.

Response 6: Thanks for your good suggestions. According to your suggestion, the line 236 has been revised in the manuscript. Red font parts were the result of modified.

Comments 7: Line 237 method referred to a previous study – what kind of wastewater?

Response 7: Thanks for your good suggestions. According to your suggestion, the line 237 has been revised in the manuscript. Red font parts were the result of modified.

Comments 8: Point 2.7 line 234-255 - one form of presentation should be maintained: “was added….”, “was cooled…..” and then: “take 15 mL of the ethyl 242 acetate layer and mix it with …….” – choose one form proper for the journal and be consistent and start a new sentence with a capital letter.

Response 8: Thanks for your good suggestions. According to the suggestion, one form of presentation have been maintained. A capital letter have been used in a new sentence. The detection method of theaflavins, thearubigins, theobromines in section 2.7 has been rewritten. Red font parts were the result of modified.

Reviewer 3 Report

Comments and Suggestions for Authors

The manuscript appear interesting but for its publication it needs some checks and changes. Text should be revised and repetition reduced. Also the lexicon must be improved, finally the descriptions of data that are present (or could be) in tables must be reduced and  discussion about reasons for the differences must be hypothesized.

Introduction

The introduction part should be check, some aspects are repeated, for example the issues about black tea and its flavour, taste  and biological function.

Line 40. For my ignorance I do not understand what “mu” is. Could authors report the entire name?

Line 44. For better understanding by readers, could authors report the value also in euro?

Lines 80-81. Theaflavin is repeated several times, in this part and in others (see other comment), Authors should check it.

Material and methods

2.5. Processing of Fruity Black Tea

Authors should report how long the fermentation phase lasted, indicating the quality parameters that have been checked on tea leaves. This helps the understanding of the process. Also the duration of the final drying operation should be explain. Is it carried out in a standardise mode? It could be interesting to understand if some active compounds could be loss.

Line 139. I am sorry but I do not know what gross and foot flame are. Could authors explain me?

2.4. Sensory evaluation of fruity black tea

Lines 149-159. Is the percentage system used codified by China National Standard (GB/T 23776–2018)? I am unable to consult this rule.

2.5. Inhibition of α-amylase activity by fruity black tea samples

Is correct used the he word tea  repeated?

2.6. Determination of water extracts, total polyphenols, total amino acids, caffeine and soluble 175 sugar

Line 213. What the letter “R” is? Probably a typing mistake?

Line 220-223. Is a calibration curve used to determine the sugar amount? Authors should explain the quantification method.

Lines 232-233. The quantification method for caffeine is not reported. Authors should describe it.

2.7. Determination of theaflavins, thearubigins, theobromines of fruit black tea

Is the reported previous study published? If yes, the reference should be inserted into text or the note of “unpublished results” should.

Lines 235-237. Check text, a repletion occurs.

Results

Lines 270-282. This part appears as a sequel of repetitions. They described data reported in table 1, I suggest to simplify and underlined the most important aspects and not report the significantly score’s differences (higher) between to samples.

Line 285. I am sorry but I do not know what the word “urun” mean. Could authors explain me, please?

Lines 284-308. The descriptions of the different teas are very repetitive. Probably, if the attributes for the sensory quality are reported into a table (like table 1)  the observation of this manuscript’s part could be easier.

Line 355. This part is not clear. Authors should check it.

Lines 390-392-393-394-397-427. Theaflavin is repeated several times, it is not clear why.

Figure 1. Check the figure’s caption symbols reported in it do not coincide with samples names reported in graph.

Figure 2. The caption should explain the letters’ mean. I suppose the differences were evaluated separately for samples before and after fermentation. From figure’s caption is not clear, authors should briefly reported in it.

Conclusion

Line 456. Authors reported the pheophytin content. But in the other parts of the text (materials and methods and results) this component has never been reported. Could authors give some explanation.

Author Response

Comments 1: The manuscript appear interesting but for its publication it needs some checks and changes. Text should be revised and repetition reduced. Also the lexicon must be improved, finally the descriptions of data that are present (or could be) in tables must be reduced and  discussion about reasons for the differences must be hypothesized. 

Response 1: Thanks for your good suggestions. According to your suggestion, text have been revised and repetition have been reduced. The lexicon also have been improved. The descriptions of data in tables have been reduced. The discussion about reasons for the differences have been hypothesized. Red font parts were the result of modified.

Comments 2: The introduction part should be check, some aspects are repeated, for example the issues about black tea and its flavour, taste and biological function.

Response 2: Thanks for your good suggestions. According to your suggestion, the introduction part have been checked and revised, which could be seen in line 70-75, 88-90. The repeated content have been revised. Red font parts were the result of modified.

Comments 3: Line 40. For my ignorance I do not understand what “mu” is. Could authors report the entire name?

Response 3: Thanks for your good suggestions. I'm so sorry. This is a mistranslation. It has been corrected. Red font parts were the result of modified.

Comments 4: Line 44. For better understanding by readers, could authors report the value also in euro?  

Response 4: Thank you very much for your good suggestions. According to your suggestion, the value have been revised in euro. Red font parts were the result of modified.

Comments 5: Lines 80-81. Theaflavin is repeated several times, in this part and in others (see other comment), Authors should check it.

Response 5: Thanks for your good suggestions. According to your suggestion, the repeated ‘Theaflavin’ have been deleted. Red font parts were the result of modified.

Comments 6: Authors should report how long the fermentation phase lasted, indicating the quality parameters that have been checked on tea leaves. This helps the understanding of the process. Also the duration of the final drying operation should be explain. Is it carried out in a standardise mode? It could be interesting to understand if some active compounds could be loss.

Response 6: Thanks for your good suggestions. According to your suggestion, the parameters of withering, fermentation, rolling, drying have been added, which could be seen in line 132-136. Red font parts were the result of modified.

Comments 7: Lines 149-159. Is the percentage system used codified by China National Standard (GB/T 23776–2018)? I am unable to consult this rule.

Response 7: Sorry to know that you are unable to consult this rule. The percentage system in manuscript was indeed codified by China National Standard (GB/T 23776–2018). The rule was issued by China General Administration of Quality Supervision, Inspection and Quarantine. Red font parts were the result of modified.

Comments 8: Is correct used the he word tea  repeated?

Response 8: Thanks for your good suggestions. According to the suggestion, the tea  repeated word have been deleted.

Comments 9: Line 213. What the letter “R” is? Probably a typing mistake?

Response 9: Thanks for your good suggestions. According to the suggestion, “R” has been deleted. The sentence in line 213-230 have been revised. Red font parts were the result of modified.

Comments 10: Line 220-223. Is a calibration curve used to determine the sugar amount? Authors should explain the quantification method.

Response 10: Thanks for your good suggestions. A calibration curve indeed had been used to determine the sugar amount. According to your suggestion, the calibration curve had been added in manuscript. Red font parts were the result of modified.

Comments 11:Lines 232-233. The quantification method for caffeine is not reported. Authors should describe it.

Response 11: Thanks for your good suggestions. According to your suggestion, The quantification method for caffeine had been added. Red font parts were the result of modified.

Comments 12:Is the reported previous study published? If yes, the reference should be inserted into text or the note of “unpublished results” should. Lines 235-237. Check text, a repletion occurs.

Response 12: Thanks for your good suggestions. According to the suggestion, the Chinese national standard “Determination of water extract content” (GB/T 8313-2018) have been added. Lines 235-237 was not a repletion occurs. Red font parts were the result of modified.

Comments 13:Lines 270-282. This part appears as a sequel of repetitions. They described data reported in table 1, I suggest to simplify and underlined the most important aspects and not report the significantly score’s differences (higher) between to samples.

Response 13: Thanks for your good suggestions. According to the suggestion, the key important aspects had been simplified and underlined. Red font parts were the result of modified.

Comments 14:Line 285. I am sorry but I do not know what the word “urun” mean. Could authors explain me, please?

Response 14: Thanks for your good suggestions. “urun” was mistake. It had been deleted.

Comments 15: Lines 284-308. The descriptions of the different teas are very repetitive. Probably, if the attributes for the sensory quality are reported into a table (like table 1)  the observation of this manuscript’s part could be easier.

Response 15: Thanks for your good suggestions. According to your suggestion, the descriptions of the different teas have been added into table 1. Red font parts were the result of modified.

Comments 16: Line 355. This part is not clear. Authors should check it.

Response 16: Thanks for your good suggestions.According to your suggestion, This part have been revised and improved. Red font parts were the result of modified.

Comments 17: Lines 390-392-393-394-397-427. Theaflavin is repeated several times, it is not clear why.

Response 17: Thanks for your good suggestions. According to your suggestion, the repeated Theaflavin have been deleted. Red font parts were the result of modified.

Comments 18: Figure 1. Check the figure’s caption symbols reported in it do not coincide with samples names reported in graph.

Response 18: Thanks for your good suggestions. According to your suggestion, the the figure’s caption symbols have been revised. Red font parts were the result of modified.

Comments 19: Figure 2. The caption should explain the letters’ mean.the letters’ mean I suppose the differences were evaluated separately for samples before and after fermentation. From figure’s caption is not clear, authors should briefly reported in it.

Response 19: Thanks for your good suggestions. The lowercase letters a-c indicate the significance of the difference. According to your suggestion, the Figure 2 have been improved. Red font parts were the result of modified.

Comments 20: Line 456. Authors reported the pheophytin content. But in the other parts of the text (materials and methods and results) this component has never been reported. Could authors give some explanation.

Response 20: Thanks for your good suggestions. The pheophytin had been deleted.

Round 2

Reviewer 3 Report

Comments and Suggestions for Authors

The manuscript has been improved. Some aspects should be more define respect to the revision of round 1. The lines’ numbers are those of the last version.

Line 46. In my comment of the previous manuscript’s version, I have asked authors to report value also in euro value.  Probably authors have forgotten.

Line 169. I have asked, in the previous revision, if it is correct to use the word tea repeated. Authors have answered that repeated word tea have been deleted. This is not what appears from this manuscript’s version.

At lines 384, 385, 386 and 387, theaflavin is repeated several times, it is not clear why. I have carried out this observation in my previously revision of the manuscript. In their “response 17”, authors have claimed: “ the repeated theaflavin have been deleted.”. I suggest to check better.

Author Response

Comments 1: Line 46. In my comment of the previous manuscript’s version, I have asked authors to report value also in euro value.  Probably authors have forgotten. 

Response 1: Thanks for your good suggestions. According to your suggestion, the report value has been expressed in euro value. Red font parts were the result of modified. In fact, this had been revised in another manuscript. However, this version of the manuscript had not been uploaded. I'm apologize for the inconvenience. 

Comments 2: Line 169. I have asked, in the previous revision, if it is correct to use the word tea repeated. Authors have answered that repeated word tea have been deleted. This is not what appears from this manuscript’s version.

Response 2: Thanks for your good suggestions. According to your suggestion, the repeated word ‘tea’ has been deleted. Red font parts were the result of modified. In fact, this content had been revised in another manuscript. However, this version of the manuscript had not been uploaded. I'm apologize for the inconvenience.

Comments 3: At lines 384, 385, 386 and 387, theaflavin is repeated several times, it is not clear why. I have carried out this observation in my previously revision of the manuscript. In their “response 17”, authors have claimed: “ the repeated theaflavin have been deleted.”. I suggest to check better.

Response 3: Thanks for your good suggestions. According to your suggestion, the repeated word ‘theaflavin’ has been deleted. The content of the statement in lines 378, 379, 380, 385, 384, 385, 386, 387, 390, 391, 392, 396, 397, 400, 401, and 402 had also been revised and refined. Red font parts were the result of modified. In fact, these contents had been revised in another manuscript. However, this version of the manuscript had not been uploaded. I'm apologize for the inconvenience.